# A Study on the Effect of Contact Pressure during Physical Activity on Photoplethysmographic Heart Rate Measurements

**DOI:** 10.3390/s20185052

**Published:** 2020-09-05

**Authors:** Francesco Scardulla, Leonardo D’Acquisto, Raffaele Colombarini, Sijung Hu, Salvatore Pasta, Diego Bellavia

**Affiliations:** 1Department of Engineering, University of Palermo, 90128 Palermo, Italy; leonardo.dacquisto@unipa.it (L.D.); raffaele.colombarini@community.unipa.it (R.C.); salvatore.pasta@unipa.it (S.P.); 2Wolfson School of Mechanical, Electrical and Manufacturing Engineering, Loughborough University, Loughborough, Leicestershire LE11 3TU, UK; S.Hu@lboro.ac.uk; 3Department for the Treatment and Study of Cardiothoracic Diseases and Cardiothoracic Transplantation, IRCCS-ISMETT, via Tricomi n.5, 90127 Palermo, Italy; dbellavia@ismett.edu

**Keywords:** photoplethysmography (PPG), PPG sensor, wearables, contact pressure, contact force, heart rate (HR), heart rate signal, PPG accuracy, heart rate reliability

## Abstract

Heart rate (HR) as an important physiological indicator could properly describe global subject’s physical status. Photoplethysmographic (PPG) sensors are catching on in field of wearable sensors, combining the advantages in costs, weight and size. Nevertheless, accuracy in HR readings is unreliable specifically during physical activity. Among several identified sources that affect PPG recording, contact pressure (CP) between the PPG sensor and skin greatly influences the signals. Methods: In this study, the accuracy of HR measurements of a PPG sensor at different CP was investigated when compared with a commercial ECG-based chest strap used as a test control, with the aim of determining the optimal CP to produce a reliable signal during physical activity. Seventeen subjects were enrolled for the study to perform a physical activity at three different rates repeated at three different contact pressures of the PPG-based wristband. Results: The results show that the CP of 54 mmHg provides the most accurate outcome with a Pearson correlation coefficient ranging from 0.81 to 0.95 and a mean average percentage error ranging from 3.8% to 2.4%, based on the physical activity rate. Conclusion: Authors found that changes in the CP have greater effects on PPG-HR signal quality than those deriving from the intensity of the physical activity and specifically, the individual best CP for each subject provided reliable HR measurements even for a high intensity of physical exercise with a Bland–Altman plot within ±11 bpm. Although future studies on a larger cohort of subjects are still needed, this study could contribute a profitable indication to enhance accuracy of PPG-based wearable devices.

## 1. Introduction

Wearable and portable technologies have recently spread in everyday life [1,2,3], and this trend is expected to reach higher percentages in the next years. The explosion of such interest in wearable technologies depends on their potential to provide continuous physiological information in real-time via an affordable and noninvasive device for health applications such as the monitoring of chronic diseases and aging populations [4,5].

Moreover, thanks to recent advances in technology and in miniaturized chips, it is now possible for a single wearable device to integrate a wide range of different sensors [6] to health awareness. 

The high request from the market for innovative wrist-worn devices is moving many companies in introducing new functionalities and new sensors in their devices, and continuous academic research concerning wearable sensors has been carried forward during the last 10 years [5,7,8]; the expected result is a continuous increase in the quality and quantity of functionalities and reliability of wearable devices aimed at enhancing the quality of life through the monitoring and maintenance of personal physiological parameters. 

Among different technologies, optical sensors based on photoplethysmography (PPG) have become increasingly popular [9,10] and nowadays have been adopted and integrated in wrist-worn products of several companies. PPG technology relies on a light emitting diode and a photo detector that is able to monitor changes in the light intensity, which are associated with changes in perfusion of the portion of tissue underneath the sensor. 

The PPG signal is composed of two components, a direct current (DC) component and an alternating component (AC). The DC component depends on the characteristics of the tissues and on the average of blood volume, and it maintains a constant absorption characteristic during the measurement. The AC component reflects the changes in blood volume and can be extracted from the DC component, to which it is superimposed, and used to calculate physiological parameters, such as the heart rate (HR). Since its amplitude is only 1% to 2% of that of the DC component [11], it is susceptible to the presence of movements and electrical noises. Indeed, one of the major issues in using PPG-based wrist-worn devices is their poor capability in tracking a reliable PPG signal during daily routine activities and physical exercise due to motion artifacts caused by hand movements [12]. 

Despite the susceptibility to motion artifacts, PPG technology is able to provide several physiological parameters associated to the cardiovascular system, such as HR, heart rate variability [13,14,15], blood oxygenation saturation [11,16,17], blood pressure [18,19,20] and arterial stiffness [21,22]. In particular, HR is one of the first parameters observed in order to monitor subject’s health in a wide range of situations, such as patient monitoring [23], training of athletes [24] and workers’ safeness [25]. 

The electrocardiogram (ECG) has been used for many years as the principal HR monitoring technology. Although this technology offers a very good accuracy [26], it does not offer sufficient user portability. In comparison, PPG-based wristband-type device are wearable and comfortable, still providing reliable cardiovascular parameters thanks to the improvements that are continuously proposed [27,28,29,30,31,32,33,34], representing an excellent potential solution to replace ECG in monitoring cardiovascular parameters during daily activities.

To fully exploit this potential, it is essential that the accuracy of PPG sensor [35] remains adequate even during physical activity and in free living conditions. Among several sources that affect PPG recording (e.g., measurement site, specific biological and physiological characteristics of the subject, breathing, ambient light and temperature), contact force between the sensor and the skin greatly influences the PPG signal [36,37,38,39,40,41] and causes motion artefacts, which are known to be a limiting factor that prevents the straight-forward usage of PPG and are considered to result from sensor-tissue motion and internal tissue movement [42]. Specifically, contact force influences both the relative motion between the sensor and the measuring site (especially during physical activities) and the arterial geometry which can be deformed by compression [43]. The contact force between the sensor and the skin can alter the subcutaneous perfusion, and eventually, it can obstruct microcirculatory blood flow. These perturbations and alterations lead to a distortion of the peak points of the PPG waveform and can lead to errors in detecting and calculating the HR, limiting all the practical applications of PPG-based devices in physiological monitoring. With an increasing contact pressure between the sensor and the skin (Figure 1), the DC amplitude increases, whereas the AC component increases first, with a maximum in the range of the optimal contact pressure, and then it decreases to the point that it is no longer possible to recognize the pulsation. 

The best PPG signal with the highest amplitude, can theoretically be obtained under conditions of transmural pressure, defined as the pressure difference between the inside and outside of the blood vessel [9]. Consequently, insufficient or excessive contact pressure (CP) leads to low signal amplitude, a poor signal to noise ratio, as well as distorted waveforms. Santos and colleagues [44] developed a stand-alone pulse oximeter based on PPG technology and equipped with a contact force sensor and found that contact force has influences also in SpO_2_, which was found to decrease as the CP was increasing. Nevertheless, assessing the optimal range of contact force is still challenging due to the wide variability of the subjects in terms of age, gender and arterial stiffness [43,45]. 

Although the effect of contact force in PPG signal quality has been investigated mainly during resting conditions, a deeper knowledge of this interaction is needed during physical activity and from a measurement site, which is widely used by commercial products (i.e., wrist), so that it is possible to take into account all the different potential sources which affect signals in a likely daily life case scenario. With this aim, in this study we considered a PPG device in reflectance mode operation, as this approach is more likely to provide comfort and more daily usability for end users.

An integrated PPG-based system was developed for measuring the HR at different CP between the sensor and the skin to determine the quality of the PPG based HR measurement at different physical activities over a cohort of seventeen subjects. By applying three different levels of CP (i.e., 12, 33 and 54 mmHg; the different acquisitions shown in Figure 1 were acquired before performing the experimental protocol in order to identify the target range of CP), the goal was to assess the optimal values of CPs to achieve a reliable estimation of HR measurements during different physical activities. 

## 2. Materials and Methods

The objective of the study is to assess the HR acquired through a PPG sensor during physical activity from the wrists of participants at different tightening pressures. The reference system adopted in the study to compare the acquisitions was a commercial ECG-based chest strap. 

To estimate how close the PPG-HR is to the ECG values, data analysis was performed through the Pearson correlation coefficient (r), the mean-average-percentage-error (MAPE) and the Bland–Altman diagram.

The measurement devices, the protocol for human subjects and the data analysis are described below:

### 2.1. Measurement Device

Two different devices were used to collect simultaneously an ECG R-R interval from the chest and a PPG waveform from the wrist of participants who took part in the protocol. The ECG data, as a reference, were collected by using a Polar chest strap (H9, Polar Electro, Kempele, Finland) and recorded via a mobile app (Elite HRV Inc., Asheville, NC, USA) which provides an array containing all the R-R intervals.

The PPG data were collected by using a custom made wrist-wearable device shown in Figure 2. 

The PPG system consists of: (i) a commercial PPG sensor (DFRobot, Shanghai, China) equipped with a wavelength of 520 nm, (ii) a 3 axis ADXL 345 accelerometer (Sparkfun Electronics, Niwot, CO, USA), (iii) a load button cell (FX1901, Meas. Spec., Schaffhausen, Switzerland), (iv) and a polylactide frame created by means of the fused filament fabrication where all the components are mounted together. 

As the load cell was placed right above the PPG sensor that allowed the quantification of the CP between the skin and the PPG sensor (contact area = 473 mm^2^, a value close to that of many commercial products) in order to investigate its influence on the reliability of the HR acquisitions during physical activity. Specifically, three different contact pressures were identified (i.e., CP1 = 12 mmHg, CP2 = 33 mmHg and CP3 = 54 mmHg) which corresponded to the three tightening levels commonly used by smartwatch users. The fixed contact area, which is intentionally close to the surface of most PPG devices on the market, prevented us from reaching higher contact pressures than 54 mmHg without causing discomfort to the users. Even if higher contact pressures could have been achieved by reducing the contact surface, it would have no longer reflected the surfaces of commercial products and would, therefore, be out of line with the main purpose of the study. Therefore, 54 mmHg tightening pressure represents, according to the point of view of the participant to the tests, the boundary level of affordable contact pressure that can be reached in most commercial products without disturbing the users.

The accelerometer, which is exposed to the same acceleration of the PPG sensor, was used to monitor the orientation of the arm over subjects and the effective rate of the exercise. 

The system was finally fastened on the wrist by using a 20 mm wide nylon strap, as shown in Figure 3. All wires were then fixed onto the arm to prevent any unwanted displacement of the system during physical activity.

All data were acquired by using an Arduino Mega 2560 board (Arduino, Turin, Italy), recorded with a sampling frequency of 250 Hz and then analyzed off-line using a dedicated Matlab^®^ algorithm (MathWorks, Inc., Natick, MA, USA).

### 2.2. Human Study Protocol

The human study protocol used in this work were approved by the Institutional Research Review Board of the institute IRCCS-ISMETT and it has been classified as IRRB/29/20. The tests were carried out between late spring and early summer 2020 in compliance with the national covid-19 secure guidance.

Seventeen subjects ranging from 22 to 55 years old (Male = 12, Female = 5; age = 36 ± 11 years, height = 173 ± 6 cm, weight = 73 ± 9 kg) were enrolled for the experimental protocol and all subjects gave their informed consent for inclusion before they participated in the study. None of them took stimulants or drugs before the tests that could have influenced HR variations. Sixteen subjects presented a skin color classification of Type II, based on the Fitzpatrick scale, and one subject presented a skin classification of Type III. As the skin pigmentation affects the PPG signal [33,46,47], a sample of subjects with the same color was selected to reduce the influence of this parameter. 

Subjects were asked to wear simultaneously the Polar H9 chest strap and the prototypal wristband and stand still in front of a 22.5 cm high step. After the study coordinator settled the wristband to one of the three predefined CPs, participant went up and down the step for 60 s at three different intensities of physical activity (i.e., low = 90 bpm, medium = 120 bpm and high = 140 bpm) for a total of 9 exercise sequences for each subject (3 tightening levels of the wrist bracelet at three different activity intensities executed in a random order). A metronome was used as a guide for participants during each exercise, and 10 s of signal at rest were recorded before the execution of each physical activity. The CP data were monitored both online and offline at the start and end of the test to ensure that the chosen CP had not changed during the physical activity.

### 2.3. Data Analysis

ECG and PPG data were processed offline with Matlab^®^, and in Figure 4, it is possible to observe the principal steps of the signal conditioning. To reduce noise, a low pass filter with a cutoff frequency of 5 Hz was applied to the digitized PPG waveform as well as a Hampel filter to detect and remove any outliers, as it has been already tested as an effective algorithm for the detection and removal of false peaks [48,49]. 

The two datasets (ECG and PPG) were synchronized by means of minimum bpm error by using the 10 s rest-condition signal to find starting points with maximum correlation values [24], and then, a PPG signal was used to extract waveform features as the peaks.

The HR was quantified both for every single pair of consecutive peaks and by using a moving average from a variable subset of data ranging linearly from 5 at 60 bpm up to 13 at 150 bpm.

ECG and PPG derived HR were finally compared (17 subjects, 9 different tests for each person), and the parameters of interest were assessed.

## 3. Results

The data from the three axis accelerometer showed no substantial differences between subjects at each physical activity. The mean and standard deviation values of the acceleration peaks during the test were 0.34 ± 0.08 g and 0.21 ± 0.04 g, respectively, in the *Y*-axis and *X*-axis (i.e., longitudinal and sagittal axis). Moreover, a qualitative inspection of the acceleration data in the frequency domain confirmed that all subjects respected the physical rhythm agreed for each test.

Similarly, data acquired from the load cell varied slightly during the execution of the test but returned to their initial value at the end of the physical activity, demonstrating that all the variations were due to muscular movements while the buckle stayed stuck in place without loosening up. Synchronized PPG and ECG signals recorded from a single subject which reflect the averaged results are shown, as example, in Figure 5.

From a qualitative point of view, it is possible to observe that for a contact pressure of 12 mmHg (CP1), the PPG signal did not follow the trend of ECG signal and this was found to be true for almost all subjects. It is also possible to observe that at 54 mmHg (CP3), the PPG signal followed the trend very well, tracing even the small variations in heart rate detected by the ECG chests trap.

To establish the accuracy of PPG-based device with respect to the gold standard (i.e., ECG device) the Pearson Correlation coefficient and the MAPE were calculated for all subjects, which are shown in Table 1 and Table 2.

It is possible to observe, as expected, that for every contact pressure, the Pearson coefficient decreased for a higher rate of the physical exercise, while the MAPE increased. The comparison of PPG and ECG-based devices was further tested by using the Bland–Altman technique [50] that has been widely used [25,51,52] to evaluate physiological parameters, and the results are show in Figure 6. 

As it is possible to observe from Table 2, the MAPE was particularly low at CP3, the maximum contact pressure (i.e., 54 mmHg), which provided the best results for any exercise rate and represented the optimal CP when taking into account all the subjects. However, it is worth noting that the standard deviation associated with the MAPE suggests that the optimal CP depended on the specific characteristics of each participant of the study due to the subjective variability. Indeed, the actual force exerted at the artery wall would be different for each subject since the arterial depth may vary from subject to subject. This, together with the thickness of the fat layer, the hydration and the specific characteristics of the biological tissues, including the skin color, may contribute to the intersubject variability in the recording of the HR.

Subsequently, we considered the best MAPE results of every single subject, to obtain a deeper analysis of the intersubject variability. Specifically, in Table 3 it is possible to observe, for each intensity of physical exercise, the number of subjects who presented their individual lowest MAPE as the contact pressure changed. For instance, for an exercise rate of 90 bpm, Subject 3 presented a MAPE of 8.54%, 1.34% and 4.54%, respectively, for a contact pressure of 12, 33 and 54 mmHg. Therefore, this subject fell within the category of the 9 subjects who presented their individual optimal contact pressure at 33 mmHg.

It is possible to observe from Table 3 that for the lowest CP, only one subject presented their lowest MAPE at a physical activity of 120 bpm and no one at an intensity of 90 and 140 bpm. At the two higher physical activities (i.e., 120 and 140 bpm), the majority of subjects presented the lowest personal MAPE at a CP of 54 mmHg. Only for the low level of physical activity, a uniformity of results was observed, in which almost an equivalent number of subjects presented the lowest MAPE at the contact pressures of 33 and 54 mmHg. A further insight into the overall results presented in Table 1, by considering recorded data at a fixed CP and physical activity level for each single subject, shows that residual data distribution approached a normal one as the CP condition approached the optimal CP level found for each subject as summarized in Table 3.

Then, we created a subset of these data (i.e., best individual results), assessing a Bland Altman plot, showed in Figure 7. We further compared the MAPE and Bland Altman of this subset (Table 4) with the best results of the whole dataset (i.e., 54 mmHg). From the comparison between the optimal contact pressure and the best results of the whole dataset (i.e., 54 mmHg), it is possible to notice that the MAPE decreased −47% for the low level of physical activity, −23% for the medium and −38% for the high level of physical activity.

Finally, no statistical significance was found focusing the analysis on the body mass index (BMI) or on the circumference of the wrist of the population sample examined. A comparison was assessed using Bland–Altman analysis between the 5 female subjects and 5 male subjects randomly chosen. Observed average standard deviation in the female sample was 48% lower. However, given the limited number of female subjects, this analysis has no statistical relevance and detailed results are not reported.

## 4. Discussion

The current study investigated the influence of CP in the reliability of the PPG-based device for HR evaluation during different intensities of physical activity. The gold standard device for the comparison was a Polar ECG-based chest strap, which was found to have a good validity during body movements in previous studies [25,52]. 

To the best of the authors’ knowledge, in the literature, there are studies which conducted CP tests with a PPG sensor placed on the fingers or in any case during static experiments [36,37,39,40,43], but an extremely limited number of them focused on the relation between physical activity and CP in PPG-based measurements [53]. However, during static tests, it is not possible to observe all the possible influencing quantities acting during physical activity. The purpose of this study was to evaluate the influence of CP on HR measurements acquired at different physical exercise rates on a sample of 17 subjects. The three contact pressures, which are dependent on the contact area between the sensor and the skin (i.e., 473 mm^2^), were chosen based on the three most common intensities with which users use the smartwatch. Specifically, the three contact pressures were selected asking five smartwatch users to wear the bracelet in a comfortable, normal and narrow position in order to quantify the contact pressures.

The authors found that CP between the PPG sensor and the skin influenced the signal recorded on the wrist of the participants who took part in the experiment. Specifically, results have demonstrated that different contact pressures provide significant differences in signal quality and reliability. While the PPG-HR at a low CP (i.e., 12 mmHg) showed a very weak correlation with ECG-HR, in accordance with our previous study [41], CP2 and CP3 (i.e., 33 and 54 mmHg respectively) provided the most accurate results. Although the difference between the two major contact pressures is not substantial, both in terms of Pearson’s correlation coefficient and MAPE, not meaningful for sport applications, it can be potentially significant for clinical purposes, where different applications need different accuracy levels both for long and short term monitoring.

In previous HR comparison tests, the results have been regarded as reliable as MAPE remains under 5% [24,25,54], and Pearson correlation coefficient ranges from 0.7 ≤ r ≤ 0.9 for a very large correlation and r > 0.9 for an excellent correlation [55,56]; Hwang and colleagues [25] investigated the accuracy of a PPG sensor embedded in a wristband-type tracker to be used by construction workers. They concluded that the PPG-based HR monitoring system has a potential to be applied at construction sites for monitoring construction workers’ HR on a real-time basis, as it showed a MAPE of 4.79% and a Pearson correlation coefficient of 0.85. However, authors specify that the accuracy needs to be further improved, particularly during heavy works. 

Our findings showed that the MAPE (Table 2) ranged from 2.4% to 4.6% for a CP of 33 mmHg and from 2.4% to 3.8% at 54 mmHg. It is worth noting that, at a fixed CP, MAPE increased as the intensity of physical activity increased, as expected. Similarly, we found a Pearson coefficient (Table 1) ranging from 0.76 to 0.93 for a CP of 33 mmHg and from 0.81 to 0.95 for a CP of 54 mmHg. 

The PPG-based HR measurements comparison was further tested by using the Bland–Altman analysis (Figure 6), which has been intensively used in wearable device performance assessments [50,57]. In a physiological monitoring study, Gatti and colleagues [55], based on previous sport studies, selected a maximum acceptable limit of agreements (LoA) in the range of ±11 bpm for HR. 

In another study concerning device accuracy on HR measurements, Lee et al. [51] considered a LoA in the range of ±11.5 as accurate and a LoA in the range of ±13.8 as less accurate. In our study, the Bland–Altman plot (Figure 6) showed the best results for the entire sample at a CP of 54 mmHg. LoA settled below ±10 bpm for the low and medium physical activity rate and in a less accurate range of ±16 bpm for the high intensity of physical exercises.

The measurement data suggest that the best results of the whole datasets (Table 3) are achieved for a CP of 54 mmHg. However, the MAPE (Table 2) indicates that among different individuals and task intensities there is a large individual-dependent variability which makes it difficult to assess a single optimal CP for the whole sample at each physical activity. However, even if 54 mmHg provided the best results among all subjects, from our study we cannot affirm that it corresponds to the upper maximum after which the signal begins to decrease again in amplitude (Figure 1) since higher pressures would have been uncomfortable for the subjects.

Table 4 shows that the best individual CP provides the lowest MAPE. Specifically, the Bland–Altman plot, shown in Figure 7, provided LoA within ±11 bpm (i.e., ±10.7) even for the high intensity of physical exercises, which is considered as reliable. 

Moreover, the MAPE of the whole dataset calculated at the lowest physical activity (i.e., 90 bpm) and for the lowest CP was 2.34 times higher than the MAPE obtained at the highest physical activity and for the highest CP. Therefore, based on the results of this study, there are two main considerations: (i) with an individual optimal CP, it is possible to consider the PPG-based HR measurement reliable even for high intensity of physical exercise (i.e., 140 bpm); (ii) the CP has greater effects on PPG-HR signal quality than those deriving from the intensity of the physical activity ranging from 90 to 140 bpm. Although these results are promising, they will be further deepened by future studies on a larger and more heterogeneous sample of subjects in terms of sex, BMI range and skin color. Moreover, it would be appropriate to repeat the measurements during different physical activities that produce different body movements that occur in common daily activities such as, walking, jogging and running. Plus, another limitation to the study was the reference device; although the ECG chest strap is extensively used as a reference in HR assessment for scientific studies, it is considered as a suboptimal reference method [58] introducing a potential additional error into wrist-worn wearable HR accuracy.

The individual optimal contact pressure can bring a potential benefit in terms of physiological measurements accuracy; however, it is necessary to have a system that can adapt the tightening pressure of the wrist-type device. Although the identification of the individual optimal CP for each subject requires a specific algorithm, currently under investigation by authors, the CP of 54 mmHg would improve the metrological qualities of PPG devices, especially during physical activity and during typical lifestyle activities.

Sim and colleague [36] proposed a PPG platform integrated with a thermopneumatic type regulator to regulate the contact-force during the measurement, adopting a target contact force of 0.6 N which showed the highest amplitude. Results showed a significant improvement of PPG measurement in terms of amplitude, suggesting a potential application of this approach to bio signals measurements in various field.

Despite numerous studies on the influence that CP has on the signal, no accepted standards have been adopted for PPG measurements of this parameter. Most of them analyzed the AC and AC/DC amplitudes of the reflected PPG signal [39]. Teng and colleagues [43] studied the change in pulse amplitude (AC) of the reflective PPG signals with increasing contact force, from 0.2 to 1.8 N. They found that for different subjects, the pulse amplitude peaked at different contacting forces, from 0.2 to 1.0 N, concluding that the actual force exerted at the artery wall would be different for each subject due to the variability between subjects.

On this basis, the effects of CP should be carefully examined in the design of PPG-based health monitoring devices as the careful control of it can bring a potential benefit in terms of accuracy and reliability.

## 5. Conclusions

Wearable PPG sensors have become very popular in the last decade thanks to their low cost, simplicity and huge potential in measuring important cardiovascular information.

Scientific interest has continued to find new physiological parameters beyond the pulse oximetry and HR to be measured with a PPG sensor, trying to fully exploit their potential. Although recent progress has been made in the hardware and signal processing to increase the accuracy of measurements, a reliable PPG sensor device, able to accurately detect HR signal during physical activity, has yet to be fully developed, and this limits the application of this technology in different fields. Among several sources that affect PPG signal, CP between the sensor and the skin greatly influences the PPG signal quality, compromising the overall reliability of the system and preventing its widespread use during the typical daily activities. Optimal CP could contribute to reducing motion artefacts and ensuring a good signal, and to determine it, in vivo PPG acquisitions were obtained from a cohort of seventeen subjects for different physical activity intensities.

The comparison between ECG and PPG signals showed the reliability and effectiveness of the proposed approach. Specifically, results show that the CP has greater effects on PPG-HR signal quality than those deriving from the intensity of the physical activity. Moreover, we observed that with an individual optimal CP it is possible to measure reliable HR signals even at a high exercise intensity. With a higher HR accuracy, a PPG-based HR sensor, integrated in a wristband, can be effectively used for monitoring athletes, workers and in general, for the personal health management for a safer and healthier lifestyle.

Although future studies on a larger cohort of subjects are needed to further strengthen our results, this study could contribute to enhancing PPG-based device accuracy in the monitoring of HR for an easy personal health management.

## Figures and Tables

**Figure 1 sensors-20-05052-f001:**
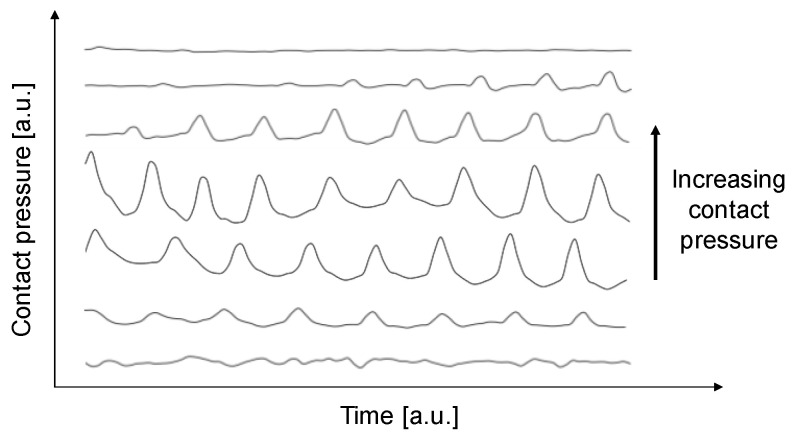
Qualitative representation of the variation of the alternating component (AC) signal for different contact pressures. With an increasing contact pressure, the amplitude initially increases up to a maximum and then it starts decreasing.

**Figure 2 sensors-20-05052-f002:**
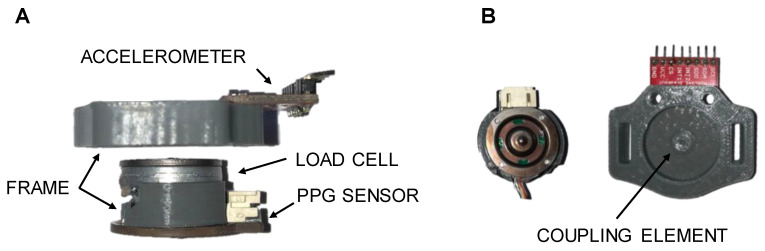
Lateral view of the prototypal measurement device configuration (**A**) and coupling elements for the load transmission (**B**).

**Figure 3 sensors-20-05052-f003:**
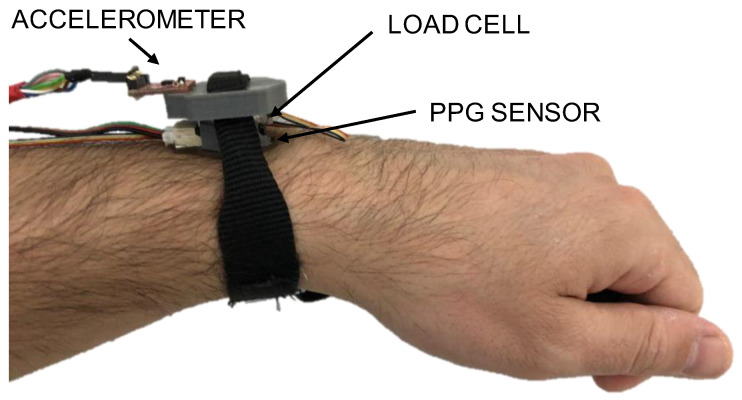
Prototypal measurement system in operation.

**Figure 4 sensors-20-05052-f004:**
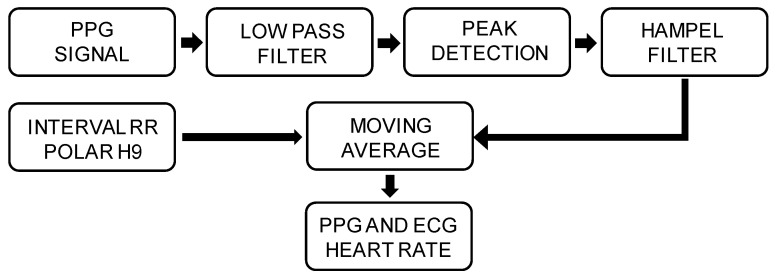
Schematic block diagram of the signal conditioning.

**Figure 5 sensors-20-05052-f005:**
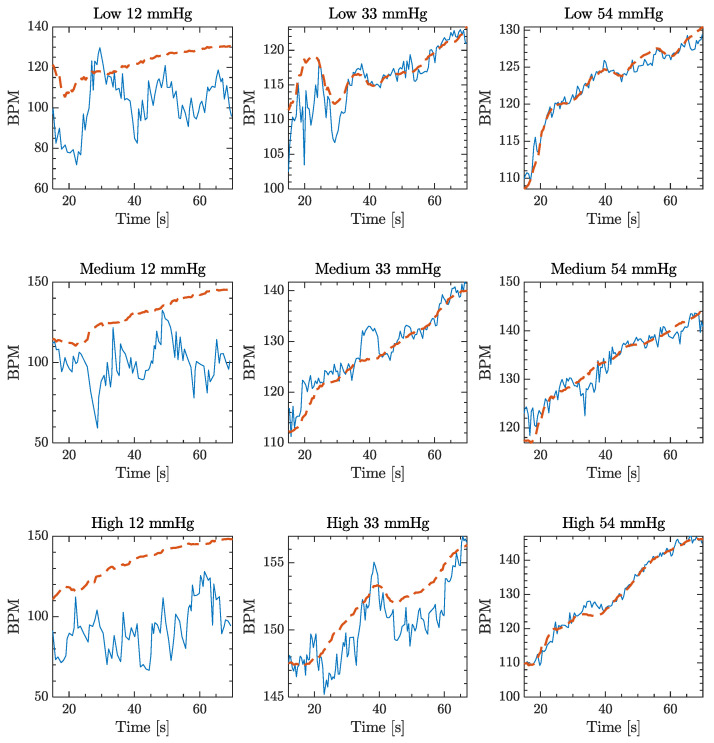
Filtered and synchronized PPG (continuous line) and ECG (dashed line) acquisitions at each physical activity and for every contact pressure (CP), to evaluate qualitatively the correlations between ECG and PPG.

**Figure 6 sensors-20-05052-f006:**
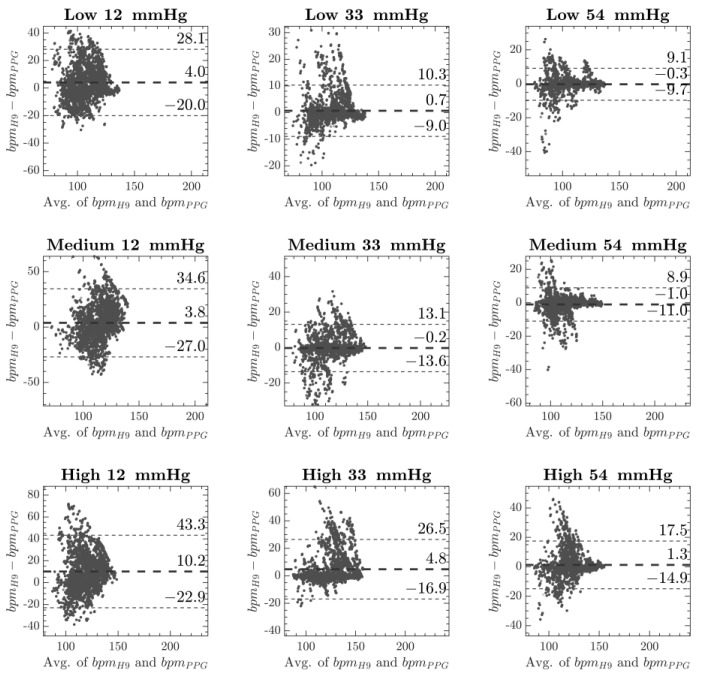
Bland Altman plot of PPG and ECG acquisition at each physical activity and for every CP, to better evaluate the correlation between ECG and PPG acquisitions.

**Figure 7 sensors-20-05052-f007:**
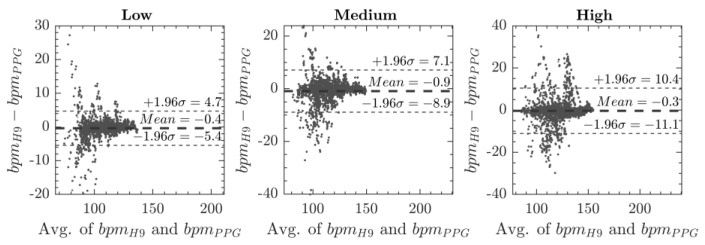
Bland Altman plot between ECG and the best individual subset of PPG acquisitions.

**Table 1 sensors-20-05052-t001:** Pearson correlation coefficient of ECG-HR and PPG-HR at each physical activity rate and for each contact pressure (i.e., CP1 = 12 mmHg, CP2 = 33 mmHg and CP3 = 54 mmHg).

	Pearson Correlation Coefficient		
Exercise Rate	CP1	CP2	CP3
90 bpm	0.56	0.93	0.95
120 bpm	0.32	0.89	0.94
140 bpm	0.28	0.76	0.81

**Table 2 sensors-20-05052-t002:** Mean average percentage error (MAPE) of ECG-HR and PPG-HR at each physical activity rate and for each contact pressure (i.e., CP1 = 12 mmHg, CP2 = 33 mmHg and CP3 = 54 mmHg).

	MAPE (σ)		
Exercise Rate	CP1	CP2	CP3
90 bpm	8.9% (4.4)	2.4% (2.7)	2.4% (3.2)
120 bpm	10.3% (4.9)	3.5% (3.5)	2.7% (3.0)
140 bpm	11.8% (6.2)	4.6% (5.3)	3.8% (3.8)

**Table 3 sensors-20-05052-t003:** Number of subjects (n) which presented the individual optimal contact pressure (Optimal CP) at every physical activity intensity.

	Lowest Individual MAPE		
Exercise Rate	CP1	CP2	CP3
90 bpm	n = 0	n = 9	n = 8
120 bpm	n = 1	n = 4	n = 12
140 bpm	n = 0	n = 6	n = 11

**Table 4 sensors-20-05052-t004:** Bland Altman and MAPE comparison between the best individual subset and whole dataset at CP3 (i.e., 54 mmHg).

	Bland–Altman Mean (± 1.96 σ)		MAPE Err % (σ)	
Exercise Rate	CP3	Optimal CP	CP3	Optimal CP
90 bpm	−0.3 (±9.4)	−0.4 (±5.0)	2.4% (3.2)	1.3% (1.9)
120 bpm	−1.0 (±10.0)	−0.9 (±8.0)	2.7% (3.0)	2.1% (2.5)
140 bpm	1.3 (±16.2)	0.3 (±10.7)	3.8% (3.8)	2.3% (2.6)

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
