# Peer review of "A Study on the Effect of Contact Pressure during Physical Activity on Photoplethysmographic Heart Rate Measurements"

_sensors, 2020, doi:10.3390/s20185052_

Round 1

Reviewer 1 Report

The manuscript presents a practical problem in the applications of the wearable PPG devices. The effect of contact pressure on the HR assessment during physical activities is described in the manuscript. The contents of the manuscript can really draw a lot of attention from the readers who work on the wearable device development. However, further modification is needed before it can be accepted and published. Some major concerns and minor errors are listed as follows:

MAJOR REVISION

  1. The authors should explain why they choose the three fixed contact pressure levels (12, 33 54 mmHg) in Discussion section.
  2. How to define the lowest individual MAPE in Table 3.
  3. In Table 4, the authors should describe how to set the optimal CP for individual subject.

MINOR REVISION

  1. “in figure no.” present at several places should be “in Figure no.”
  2. LINE 145

What is the “PLA frame“? Its full name should be used.

  1. LINE 117

”3 tightening level …” should be “3 tightening levels …”

  1. In Figure 5

In the sub-figure in the middle, row 3, of the figure, the x-axis scale is not correct.

  1. LINE 242

”every contact pressures …” should be “every contact pressure …”

  1. LINE 260

”the MAPE was …” should be “the MAPE is …”

  1. LINE 293

”MAPE decreased of -47% …” should be “MAPE decreased -47% …”

  1. LINE 344

”… of ±11,5 …” should be “… of ±11.5 …”

  1. In many lines.

”…, basing on the …” should be “…, based on the …”

  1. LINE 387

”… in different field.” should be “… in different fields.”

Author Response

We thank the reviewer for her or his valuable comments. We have taken these comments into careful consideration when preparing the revised manuscript and feel that the critiques led directly to an improved submission. We hope that the reviewer agrees. All changes made to the document were highlighted in yellow.

Reviewer #1

The manuscript presents a practical problem in the applications of the wearable PPG devices. The effect of contact pressure on the HR assessment during physical activities is described in the manuscript. The contents of the manuscript can really draw a lot of attention from the readers who work on the wearable device development. However, further modification is needed before it can be accepted and published. Some major concerns and minor errors are listed as follows:

MAJOR REVISION

  1. The authors should explain why they choose the three fixed contact pressure levels (12, 33 54 mm Hg) in Discussion section

Reply: Thank you for your comment. We agree with the reviewer that the reasons which prompted the choice of the selected contact pressures are not sufficiently clarified in the text. To clarify this aspect, the following sentence was added in the discussion section:

The three contact pressures, which are dependent on the contact area between the sensor and the skin (i.e. 473 mm2), were chosen based on the three most common intensities with which users use the smartwatch. Specifically, the three contact pressures were selected asking 5 smartwatch users to wear the bracelet in a comfortable, normal and narrow position in order to quantify the contact pressures

Moreover, thanks to the reviewer, authors recognize that it is not mentioned in the text why authors have not tested contact pressures higher than 54 mmHg, which is the best one over the entire sample of subjects. Indeed, tests with a higher contact pressure would have been desirable since we cannot affirm that a slightly higher contact pressure would have produced equally good or higher results.

In fact, Figure 1 gives a qualitative idea of ​​the amplitude trend of the signal as the contact pressure varies, which increases to a maximum and then decreases again. With the aim of finding the correct tightening level compatible with the usability of wearable devices, no contact pressures were selected that are no longer compatible with the stress of the subjects.

The contact area that we have chosen, which intentionally is comparable to the surface of most PPG devices on the market, prevented us from reaching higher contact pressures without causing complaint and inconvenience to users. In fact, during the tests, all the participants felt a condition of discomfort due to higher pressures (above 54 mmHg) and the cardiologist, who has supervised the tests, advised not to reach higher tightening pressures than 54 mmHg.

Higher pressure could have been achieved by reducing the contact surface. However, a smaller contact surface would no longer reflect the surfaces of commercial products and would therefore be out of line with the main purpose of the study.

Therefore, we can conclude that 54 mmHg represent the maximum contact pressure that can be reached in most commercial products without disturbing users.

To clarify this aspect, also the following sentences were added to the test, in the material and methods and in the discussion section, respectively:

The fixed contact area, which is intentionally close to the surface of most PPG devices on the market, prevented us from reaching higher contact pressures than 54mmHg without causing discomfort to the users. Even if higher contact pressures could have been achieved by reducing the contact surface, it would have no longer reflect the surfaces of commercial products and would therefore be out of line with the main purpose of the study. Therefore, 54 mmHg tightening pressure represents, according to the point of view of the participant to the tests, the boundary level of affordable contact pressure that can be reached in most commercial products without disturbing the users.” 

However, even if 54 mmHg provided the best results among all subjects, from our study we cannot affirm that it corresponds to the upper maximum after which the signal begins to decrease again in amplitude (Fig. 1) since higher pressures would have been uncomfortable for the subjects. “

  1. How to define the lowest individual MAPE in Table 3.

Reply: Authors thank the reviewer for her or his comment. Table 3 reports the number of subjects which presented the individual optimal contact pressure at every single physical activity. To clarify what is intended for “lowest individual MAPE” the following sentence was added in the Result section:

Specifically, in Table 3 it is possible to observe, for each intensity of physical exercise, the number of subjects who presented their individual lowest MAPE as the contact pressure changed. For instance, for an exercise rate of 90 bpm, Subject 3 presented a MAPE of 8.54%, 1.34% and 4.54% respectively for a contact pressure of 12, 33 and 54 mmHg. Therefore, this subject falls within the category of the 9 subjects who presented their individual optimal contact pressure at 33mmHg.”

Conversely, Table 4 reports the lowest individual MAPE (err % (σ)) over all subjects at each physical intensity, which is calculated for each individual contact pressure over the sample.

  1. In Table 4, the authors should describe how to set the optimal CP for individual subject.

Reply: Thank you very much for your specific comment. In this study, the best contact pressure for each subject corresponds to the individual best MAPE result obtained among the three tested contact pressures (i.e. 12, 33 and 54 mmHg).

However, these contact pressures have been identified a posteriori, after having compared them with the reference system. Authors recognize that a PPG-based system able to identify the best contact pressure a priori, without a direct comparison with a reference system, can represent a big step forward in improving the metrological quality of the PPG sensor.

The aim of the study is to provide a scientific evidence on the influence that contact pressure has on heart rate measurements made through the use of a PPG sensor. Therefore, contact pressure is a parameter that should be taken into account to enhance the metrological quality of the output of the PPG sensor, which are necessary to fully exploiting the potential of the PPG sensor for a use in a clinical-grade device.

In a commercial environment, to set the optimal CP for individual subject, a device should be equipped with a pressure sensor to provide the user with an indication of the contact pressure exerted.

Our research group is currently studying and developing an algorithm able to independently identify the optimal contact pressure for each subject without any reference device (a priori). The results will be submitted for publication as soon as the study will be completed.

The following sentence has been added to the text to clarify this aspect in the discussion section:

Although the identification of the individual optimal CP for each subject requires a specific algorithm, currently under investigation by authors, the contact pressure of 54 mmHg would improve the metrological qualities of PPG devices, especially during physical activity and during typical lifestyle activities

MINOR REVISION

  1. “in figure no.” present at several places should be “in Figure no.”

Reply: Thank you very much for your suggestion, we have changed the text accordingly.

  1. LINE 145

What is the “PLA frame“? Its full name should be used.

Reply: We agree with the reviewer that the full name should be used as PLA may be unclear. The following text was added in the Discussion section to address these comments:

“iv) a frame made from polylactide through the fused filament fabrication where all the components are mounted together.” 

  1. LINE 177

”3 tightening level …” should be “3 tightening levels …”

Reply: Thank you very much for your comment. We have fixed this comment according to the suggestion.

  1. In Figure 5

In the sub-figure in the middle, row 3, of the figure, the x-axis scale is not correct.

Reply: Thank you very much for your observation. The previous observable time shift was due to a small delay in the start of the test compared to the start of the signal acquisition. We have fixed the x-axis scale according to the suggestion.

  1. LINE 242

”every contact pressures …” should be “every contact pressure …”

  1. LINE 2 60

”the MAPE was …” should be “the MAPE is …”

  1. LINE 2 93

”MAPE decreased of -47% …” should be “MAPE decreased -47% …”

  1. LINE 344

”… of 11,5 …” should be “… of 11.5 …”

  1. In many lines.

”…, basing on the …” should be “…, based on the …”

  1. LINE 3 87

”… in different field.” should be “… in different fields.”

Reply: Thank you very much for your suggestions. We have fixed all these points.

Reviewer 2 Report

The authors present the results of an interesting study. Since heart rate is a very important physiological parameter, heart rate measurements became more and more helpful the better they can be integrated into normal everyday situations. Photoplethysmographic sensors, which are for example implemented on watches or bracelets, allow 24-hour measurements, provided that the measurements are precise and correct. The precision of the measurement depends on the pressure with which the sensor touches the skin. In the present study, measurements were performed in healthy volunteers at 3 different pressures (12, 33 and 54 mmHg). The participants performed physical movements under standardized conditions (climbing stairs). Using correlation analyses and Bland-Altmann plots, the authors conclude that a pressure of 54 mmHg provides the most accurate measurement results. ECG data with a chest strap were used as comparative values.

The paper is very well and fluently written and the issue and topic is important in view of the growing need to reliably measure vital signs under everyday conditions (e.g. physical exertion).

Abstract:

The abstract lacks a subdivision into: background, methods, results, conclusion.

Introduction:

Line 72-79: The content of this paragraph is somehow already described in previously sections and therefore the content is redundant.

Line 96-97: The authors provide a graph where they present that at low pressure the sensor signal quality is bad, at increasing pressure the signal gets better and finally at even higher pressure the signal gets worse again. Unfortunately, no information is given about which pressures are involved. However, such information is important for the further course of the paper.

Methods:

The authors do not describe which parameter they define as primary outcome.

The authors do not describe a sample size calculation.

The authors do not report any vote of an ethics committee.

The authors do not report whether the study has been registered.

The authors do not report how the participants have been informed and whether written consent forms have been submitted.

The authors do not report when the study was conducted.

The authors do not report whether, for example, coffee was taken before the measurements, which, depending on the dose, can have a corresponding effect on the heart rate.

The authors do not explain why just the 3 chosen pressures were taken and not others (e.g. even higher pressures, under which the signal quality deteriorates again).

The authors do not explain why a chest strap was used as a reference method instead of an ECG with corresponding electrodes (for example, 5 different electrodes can be used, which increases the precision of the heart rate measurement).

Results:

Pearson correlation coefficients are given, but it remains unclear whether the data were actually normally distributed or not.

The difference between 12 mmHg and the two higher pressure values is much greater than the difference between 33 and 54 mmHg (both for the correlation coefficients and for the MAPEs). It remains unclear whether the differences between 33 mmHg and 54 mgHg are significant at all.

It would have been helpful if even higher pressures had been used to illustrate that the signal quality and precision would again deteriorate (as shown in the introduction). Without using even higher pressures it remains unclear to the reader whether 54 mgHg is actually the best pressure for precise measurements.

Conclusion:

Due to the above described points it remains questionable whether the conclusions can be drawn.

Furthermore, it is questionable whether the results can be generalized, since the total number of participants is rather low and the gender distribution remains very inhomogeneous.

In summary, the introductory section is far too extensive with regards to the methods section, where elementary and important information are lacking. The authors mention too few limitations of the study.

If the above-mentioned points of criticism can be adequately addressed, the present study could make a contribution to the important question of how to optimize the measurement of vital parameters under everyday conditions.

Author Response

We thank the reviewer for her or his valuable comments. We have taken these comments into careful consideration when preparing the revised manuscript and feel that the critiques led directly to an improved submission. We hope that the reviewer agrees. All changes made to the document were highlighted in yellow.

Reviewer

The authors present the results of an interesting study. Since heart rate is a very important physiological parameter, heart rate measurements became more and more helpful the better they can be integrated into normal everyday situations. Photoplethysmographic sensors, which are for example implemented on watches or bracelets, allow 24-hour measurements, provided that the measurements are precise and correct. The precision of the measurement depends on the pressure with which the sensor touches the skin. In the present study, measurements were performed in healthy volunteers at 3 different pressures (12, 33 and 54 mmHg). The participants performed physical movements under standardized conditions (climbing stairs). Using correlation analyses and Bland-Altmann plots, the authors conclude that a pressure of 54 mmHg provides the most accurate measurement results. ECG data with a chest strap were used as comparative values.

The paper is very well and fluently written and the issue and topic is important in view of the growing need to reliably measure vital signs under everyday conditions (e.g. physical exertion).

Abstract:

The abstract lacks a subdivision into: background, methods, results, conclusion.

Reply: Thank you very much for your suggestion. We have implemented the subdivision into the Abstract section.

Introduction:

Line 72-79: The content of this paragraph is somehow already described in previously sections and therefore the content is redundant.

Reply: Thank you very much for your comment. To avoid redundant content, the indicated part has been shortened as requested by the reviewer even if not completely canceled. The authors wish to keep this few lines to let every potential reader to understand the reasons why ECG-based devices are not widespread as the PPG-based despite being more reliable. The purpose of this part is to outline the weight that the wearability of a device has on its choice by a common user who wish to monitor his physiological parameters.

Line 96-97: The authors provide a graph where they present that at low pressure the sensor signal quality is bad, at increasing pressure the signal gets better and finally at even higher pressure the signal gets worse again. Unfortunately, no information is given about which pressures are involved. However, such information is important for the further course of the paper.

Reply: Thank you very much for your comment. We fully agree with the reviewer that a quantification of contact pressure would have been useful for readers. Unfortunately, it was not possible for us to record the PPG signal for contact pressures above 54 mmHg, as we will also specify later in this document. In fact, our bracelet, embedded with a load cell and a PPG sensor, has a contact surface which did not allow us to tighten the bracelet for high pressures as this caused an unpleasant feeling to the subjects involved in the study. So that to realize Figure 1 we had to use a different PPG sensor, with a smaller contact area (also smaller than those of commercial devices), which however can’t be embedded with our load cell and therefore without being able to quantify the contact pressure.

We still decided to report the Figure 1 – aimed only to provide a global highlight of the investigated phenomenon – to qualitatively show the PPG output trend as the contact pressure increases, assuming it still provides valuable and clear information.

In addition, the response of the PPG signal at different contact pressure varies from person to person (as we report in the article, the best PPG signal with the highest amplitude, can theoretically be obtained under conditions of transmural pressure) and we have decided to keep it generic without any quantification.

For these reasons, if permitted by the reviewer, we would prefer to keep this figure which, for the above considerations, we still consider useful for the readers.

Methods:

The authors do not describe which parameter they define as primary outcome.

Reply: Thank you very much for your comment. We fully agree with the reviewer that the target parameters are not outlined in the Material and Method section, and it would improve the general comprehension of the text.

The following sentence was added at the beginning of the Materials and Methods section:

“The objective of the study is to assess the HR acquired through a PPG sensor during physical activity from the wrist of participants at different tightening pressures. The reference system adopted in the study to compare the acquisitions was a commercial ECG-based chest strap.

To estimate how close the PPG-HR is to the ECG values, data analysis was performed through the Pearson correlation coefficient (r), the mean-average-percentage-error (MAPE) and the Bland-Altman diagram.”

The authors do not describe a sample size calculation.

Reply: Thank you very much for your comment. It is generally recognized (Nelson, B.W., Low, C.A., Jacobson, N. et al. Guidelines for wrist-worn consumer wearable assessment of heart rate in biobehavioral research. npj Digit. Med. 3, 90, 2020. https://doi.org/10.1038/s41746-020-0297-4 - https://www.nature.com/articles/s41746-020-0297-4#citeas) that there is a lack of standardization of the processing procedures for the assessment of heart rate in wrist-worn devices tests (e.g. participant demographic characteristics and size).

We choose the sample size based on the standard “ANSI/CTA-2065” (https://www.cta.tech/Resources/Standards), which provides participant consideration, suggesting at least 20 participants.  Unfortunately, even if the subjects enrolled in our study were originally 20, 3 were excluded due to technical problems while wearing the ECG device: we were therefore unable to process the data of more than 17 participants.

Moreover, due to the COVID restrictions, which drastically reduced the number of people authorized to enter the laboratories and those available to perform the tests, our cohort of subjects is also not heterogeneous in terms of gender as originally estimated.

The following sentence was added in the discussion section to highlight these limitation:

Although these results are promising, they will be furtherly deepen by future studies on a larger and more heterogeneous sample of subjects in terms of sex, BMI range and skin color. Moreover, it would be appropriate to repeat the measurements during different physical activities that produce different body movements that occur in common daily activities such as, walking, jogging and running.”

The authors do not report any vote of an ethics committee.

Reply: Thank you very much for your comment. We are sorry for not reporting it before. The following sentences has been added to the text:

Ethical Statements: The human study protocols used in this work were approved by the Istitutional Research Review Board of the Institute IRCCS-ISMETT and it has been classified as IRRB/29/20

The authors do not report whether the study has been registered.

Reply: Dear Reviewer, the study has not been registered yet as Authors intended this investigation as a preliminary stage given that no similar data were available in the literature or from commercial paper. The article is intended to show to the scientific community the great impact of the accurate selection of the contact pressure also in dependence of the intensity level of the physical activity.

Authors therefore plan to register the study after a next step of research activity in which we will furtherly deepen the investigation of the influence of contact pressure together with other influencing parameters (e.g. Body Mass Index, skin tone, gender) following the suggestions of ANSI/CTA-2065 Standard for physical activity monitoring for heart rate on a larger sample of subjects.

The authors do not report how the participants have been informed and whether written consent forms have been submitted.

Reply: Thank you very much for your suggestion. All subjects gave their informed consent before the tests. Therefore, the following sentence has been added in the Method section (2.2. Human study protocol):

and all subjects gave their informed consent for inclusion before they participated in the study.

Authors want to add that data gathering has been organized in order that no participant can be identified from collected data (they are anonymized) and no patient’s initial, nor birthdate or similar appear in any image or table of the article

The authors do not report when the study was conducted.

Reply: Thank you very much for your comment. The following section has been added to the article:

Conduct of tests: The tests have been carried out between late spring and early summer 2020 in compliance with the national covid-19 secure guidance”

The authors do not report whether, for example, coffee was taken before the measurements, which, depending on the dose, can have a corresponding effect on the heart rate.

Reply: As properly evidenced by the reviewer, any kind of stimulant taken before the measurements could have an effect on the tests. Thus, during the enrolling phase, all participants were specifically requested not to take any stimulant during the scheduled day of the test and before every single session participant confirmed they hadn’t taken any kind of stimulant, including coffee.

The following sentence has been reported in the text (Methods, 2.2. Human study protocol): “None of them took stimulants or drugs before the tests that could have influenced HR variations.”, assuming it can effectively address the issue evidenced by the reviewer.

* The authors do not explain why just the 3 chosen pressures were taken and not others (e.g. even higher pressures, under which the signal quality deteriorates again).

Reply: Thank you for your comment. The answer of this comment is reported later in this document.

The authors do not explain why a chest strap was used as a reference method instead of an ECG with corresponding electrodes (for example, 5 different electrodes can be used, which increases the precision of the heart rate measurement).

Reply: Thank you for your comment. We agree with the reviewer that although several research groups are currently using chest strap device as reference methods, ECG based chest strap can be considered as a suboptimal reference method if compared with the clinical reference ECG method.

Unfortunately, a standardized validity assessment protocol for physiological signal from wearable technology adopted by the vast majority of research groups is still missing; the variety of different methods available do not allow to define clear criteria.

The ANSI/CTA-2065 Standard for physical activity monitoring for heart rate, state that “the control device can be either an ECG or an electrode based device used as a HR reference in a scientifically validated, peer reviewed research paper that addresses rest and exercise published in a relevant journal”.

For sake of wearing simplicity, we opted for a chest strap. The following sentence was added in the discussion section to highlight this potential source of error:

Plus, another limitation to the study was the reference device; despite the ECG chest strap is extensively used as reference in HR assessment for scientific studies, it is considered as suboptimal reference method [58] introducing a potential additional error into wrist-worn wearable HR accuracy”.

Results:

Pearson correlation coefficients are given, but it remains unclear whether the data were actually normally distributed or not.

Reply: Thank you very much for your stimulating comment which has suggested the Authors to perform an insight in the obtained results, evidencing that residual data distribution approaches a normal one as the CP condition get closer to the optimal one of each subject as summarized in Table 3.

Therefore, the following sentence was added in the results section:

“A further insight in the overall results presented in Table 1, by considering recorded data at a fixed CP and physical activity level for each single subject, has evidenced that residual data distribution approaches a normal one as the CP condition get closer to the optimal CP level found for each subject as summarized in Table 3.” 

The difference between 12 mmHg and the two higher pressure values is much greater than the difference between 33 and 54 mmHg (both for the correlation coefficients and for the MAPEs). It remains unclear whether the differences between 33 mmHg and 54 mgHg are significant at all.

Reply: Thank you very much for your comment. The authors carried out the tests on the three most used levels of tightening to examine the difference in HR accuracy. Indeed, as evidenced by the reviewer, the difference between 33 and 54 mmHg is not so large.

PPG sensors are integrated mainly in smartwatches and, although their use is currently almost exclusively for sports use, it is hoped that this technology can provide in the future some indications that could be useful in the clinical setting.

The difference between a tightening of 33 and 54 mmHg is approximately 1 bpm which, for sports use, is hardly significant. However, for a use in healthcare, where different applications require a different degree of accuracy in measurements, it could be significant. Although 1 bpm is small, it accumulates in the case of prolonged HR monitoring over time. Even in a clinical short monitoring time of heart rate in beat to beat applications could be significant: with a contact pressure of 33 mmHg there would remain the question whether the lost beat was not detected by the device (measurement system defect) or it is a non-generated beat by the heart (true missing beat).

In addition, it should be considered that the contact pressure is only one of the several influencing parameters that influence the measurement of heart rate and it is not easy to attribute its own sensitivity coefficient to each influence parameter.

To clarify this aspect, the following sentence was added in the discussion section:

Although the difference between the two major contact pressures is not substantial, both in terms of Pearson's correlation coefficient and MAPE, and not meaningful for sport applications, it can be potentially significant for clinical purposes, where different applications need different accuracy levels both for long and short term monitoring.”

It would have been helpful if even higher pressures had been used to illustrate that the signal quality and precision would again deteriorate (as shown in the introduction). Without using even higher pressures it remains unclear to the reader whether 54 mgHg is actually the best pressure for precise measurements.

* The authors do not explain why just the 3 chosen pressures were taken and not others (e.g. even higher pressures, under which the signal quality deteriorates again).

Reply: Thank you for your comment. Indeed, as the reviewer suggests, tests with a higher contact pressure would have been desirable since we cannot assert that a slightly higher contact pressure would have produced equally good or better results.

In fact, Figure 1 gives a qualitative idea of ​​the amplitude trend of the signal as the contact pressure varies, which increases to a maximum and then decreases again. With the aim of finding the correct tightening level compatible with the usability of wearable devices, no contact pressures were selected that are no longer compatible with the stress of the subjects.

The contact area that we have chosen, which intentionally is close to the surface of most PPG devices on the market, prevented us from reaching higher contact pressures without causing complaint and inconvenience to users. In fact, during the tests, all the participants felt a condition of discomfort due to higher pressures (above 54 mmHg) and the cardiologist, who has supervised the tests, advised not to reach higher tightening pressures than 54 mmHg.

Higher pressure could have been achieved by reducing the contact surface. However, a smaller contact surface would no longer reflect the surfaces of commercial products and would therefore be out of line with the main purpose of the study.

Therefore, we can conclude that 54 mmHg represent the maximum contact pressure that can be reached in most commercial products without disturbing users.

To clarify this aspect, the following sentence was added to the test, in the Methods and Discussion section respectively:

The fixed contact area, which is intentionally close to the surface of most PPG devices on the market, prevented us from reaching higher contact pressures than 54mmHg without causing discomfort to the users. Even if higher contact pressures could have been achieved by reducing the contact surface, it would have no longer reflect the surfaces of commercial products and would therefore be out of line with the main purpose of the study. Therefore, 54 mmHg tightening pressure represents, according to the point of view of the participant to the tests, the boundary level of affordable contact pressure that can be reached in most commercial products without disturbing the users.”

However, even if 54 mmHg provided the best results among all subjects, from our study we cannot affirm that it corresponds to the upper maximum after which the signal begins to decrease again in amplitude (Fig. 1) since higher pressures would have been uncomfortable for the subjects.

Conclusion:

Due to the above described points it remains questionable whether the conclusions can be drawn.

Dear reviewer, the authors thank you for the review and for the suggestions that have certainly raised the quality and details of the article. The authors hope that the answers provided for each point have been adequate.

Furthermore, it is questionable whether the results can be generalized, since the total number of participants is rather low and the gender distribution remains very inhomogeneous.

Dear reviewer, we are agreeing that the total number of participants is rather low and the gender distribution remains very inhomogeneous. That was due to the COVID restriction as mentioned before. For this reason, we added the following text in the discussion paragraph in which we highlight this limitation:

Although these results are promising, they will be furtherly deepen by future studies on a larger and more heterogeneous sample of subjects in terms of sex, BMI range and skin color. Moreover, it would be appropriate to repeat the measurements during different physical activities that produce different body movements that occur in common daily activities such as, walking, jogging and running. Plus, another limitation to the study was the reference device; despite the ECG chest strap is extensively used as reference in HR assessment for scientific studies, it is considered as suboptimal reference method [58] introducing a potential additional error into wrist-worn wearable HR accuracy.”

Authors recognize that the sample size is a limitation of the study and that it will be necessary a confirm from new studies on a higher and better distributed cohort of subjects. However, we still hope that this study could provide an important view of the growing need to reliably measure vital signs under everyday conditions

In summary, the introductory section is far too extensive with regards to the methods section, where elementary and important information are lacking. The authors mention too few limitations of the study.

Thank you for your comment. As suggested by the reviewer in the previous sections, the introduction section has been reduced and the Methods section appreciably extended, including that important information that were missing. Moreover, the reviewer’s concerns about the limitation of the study have been properly mentioned in the discussion section.

If the above-mentioned points of criticism can be adequately addressed, the present study could make a contribution to the important question of how to optimize the measurement of vital parameters under everyday conditions.

Authors have spent great effort in addressing, at the best of their knowledge, all mentioned points of criticism (e.g. sample size, different contact pressures, reference device, statistical issues, limitation), recognizing the valuable effort of the reviewer to address all unsaid or not effectively clear considerations or indications that could enable the reader to precisely trace the technical and scientific approach adopted in the proposed investigation.

The results obtained in the proposed paper, which are more clearly and effectively presented after having addressed the specific criticism proposed by the reviewer, in the Authors opinion can appreciably contribute to a step forward in the actual literature on heart rate wearable devices, highlighting the relevant impact of contact pressure on the quality of measured heart rate. Nevertheless, results will be furtherly refined by means of a deeper investigation performed onto a wider sample, not so easily manageable during the actual Covid emergency.

Reviewer 3 Report

In this paper, the author uses ECG signal as the gold standard to measure the heart rate accuracy of PPG signal obtained by different contact pressure under different exercise intensity scenarios. The research is interesting.

I have some concerns on the logic of experiment design and presentation.  As indicated in the paper, CP of 54 mmHg provides the most accurate outcome. Based on the experimental results, we may also seem to be able to conclude that the higher the continuous pressure, the higher the accuracy of PPG. If that's the case, 54 mmHg as the "Optimal CP" level need be re-considered. What if CP level gets higher?

Author Response

We thank the reviewer for her or his valuable comments. We have taken these comments into careful consideration when preparing the revised manuscript and feel that the critiques led directly to an improved submission. We hope that the reviewer agrees. All changes made to the document were highlighted in yellow.

Reviewer 

In this paper, the author uses ECG signal as the gold standard to measure the heart rate accuracy of PPG signal obtained by different contact pressure under different exercise intensity scenarios. The research is interesting.

I have some concerns on the logic of experiment design and presentation.  As indicated in the paper, CP of 54 mmHg provides the most accurate outcome. Based on the experimental results, we may also seem to be able to conclude that the higher the continuous pressure, the higher the accuracy of PPG. If that's the case, 54 mmHg as the "Optimal CP" level need be re-considered. What if CP level gets higher?

Reply: Thank you for your comment. Indeed, as the reviewer suggests, tests with a higher contact pressure would have been desirable since we cannot assert that a slightly higher contact pressure would have produced equally good or better results.

In fact, Figure 1 gives a qualitative idea of ​​the amplitude trend of the signal as the contact pressure varies, which increases to a maximum and then decreases again. With the aim of finding the correct tightening level compatible with the usability of wearable devices, no contact pressures were selected that are no longer compatible with the stress of the subjects.

The contact area that we have chosen, which intentionally is comparable to the surface of most PPG devices on the market, prevented us from reaching higher contact pressures without causing complaint and inconvenience to users. In fact, during the tests, all the participants felt a condition of discomfort due to higher pressures (above 54 mmHg) and the cardiologist, who has supervised the tests, advised not to reach higher tightening pressures than 54 mmHg.

Higher pressure could have been achieved by reducing the contact surface. However, a smaller contact surface would no longer reflect the surfaces of commercial products and would therefore be out of line with the main purpose of the study.

Therefore, we can conclude that 54 mmHg represent the maximum contact pressure that can be reached in most commercial products without disturbing users.

To clarify this aspect, also the following sentences were added to the test, in the material and methods and in the discussion section, respectively:

The fixed contact area, which is intentionally close to the surface of most PPG devices on the market, prevented us from reaching higher contact pressures than 54mmHg without causing discomfort to the users. Even if higher contact pressures could have been achieved by reducing the contact surface, it would have no longer reflect the surfaces of commercial products and would therefore be out of line with the main purpose of the study. Therefore, 54 mmHg tightening pressure represents, according to the point of view of the participant to the tests, the boundary level of affordable contact pressure that can be reached in most commercial products without disturbing the users.” 

However, even if 54 mmHg provided the best results among all subjects, from our study we cannot affirm that it corresponds to the upper maximum after which the signal begins to decrease again in amplitude (Fig. 1) since higher pressures would have been uncomfortable for the subjects. “

Round 2

Reviewer 1 Report

MINOR REVISION

  1. LINE 399

” …will be furtherly deepen ..” should be “…will be furtherly deepened ...”

  1. LINE 403

” …to the study was the …” should be “…to the study is the …”